# Bull’s Eye Maculopathy in Near-Infrared Reflectance as An Early Sign of Hydroxychloroquine Toxicity

**DOI:** 10.3390/diagnostics13030445

**Published:** 2023-01-26

**Authors:** Miguel Santos, Inês Leal, Tiago Morais Sarmento, Sofia Sousa Mano, Patrícia José, Sara Vaz-Pereira

**Affiliations:** 1Department of Ophthalmology, Centro Hospitalar Universitário de Lisboa Norte, EPE—Hospital de Santa Maria, 1649-035 Lisbon, Portugal; 2Department of Ophthalmology, Faculdade de Medicina, Universidade de Lisboa, 1649-028 Lisbon, Portugal; 3Instituto de Microcirurgia Ocular Lisboa, IMO Lisboa, 1600-209 Lisbon, Portugal; 4Hospital do Espírito Santo de Évora, EPE, 7000-811 Évora, Portugal

**Keywords:** bull’s eye maculopathy, diagnostic imaging, hydroxychloroquine toxicity, near-infrared reflectance imaging, optical coherence tomography

## Abstract

Hydroxychloroquine (HCQ) ocular toxicity is rare but severe, and progression can occur even after termination of therapy. Case reports have suggested that a bull’s eye maculopathy detected by near-infrared reflectance (NIR) may indicate early HCQ toxicity. This retrospective cross-sectional study evaluated patients treated with HCQ who underwent routine screening with optical coherence tomography (OCT), fundus autofluorescence (FAF) and 10-2 perimetry. NIR images captured alongside OCT were subsequently graded independently by 2 masked graders for the presence of bull’s eye maculopathy, and the result was compared to the outcome of the screening. A total of 123 participants (246 eyes) were included, and 101 (90%) were female. The patients’ mean age was 55.2 ± 13.8 years. The mean time of HCQ usage was 84.0 ± 72.3 months, and the mean weekly dose was 2327 ± 650 mg. Two eyes showed toxicity in all 3 routine screening exams, with one patient suspending HCQ. The prevalence of bull´s eye lesions in NIR was 13% (33 eyes) with substantial intergrader agreement, a 71.3% specificity and 88.0% negative predictive value for HCQ toxicity. We suggest that NIR changes may be a sign of early HCQ toxicity. The detection of NIR bull´s eye lesions may warrant an increased screening frequency.

## 1. Introduction

Hydroxychloroquine (HCQ), an antimalarial drug, is widely used for several immune-mediated diseases, the most common being systemic lupus erythematosus (SLE), rheumatoid arthritis (RA) and several dermatological conditions, such as solar urticaria and lichen planus [1,2,3,4]. Retinal toxicity from HCQ has been recognized for many years. Despite the low incidence, its impact is considerable, since many patients take these drugs for medical conditions and for long periods of time [3]. Another major concern is that even after cessation of the drug there is often no recovery of function, and sometimes the retinal condition may even progress [5]. Detection of the tell-tale sign, a bull’s eye maculopathy on fundoscopy, represents the end-stage disease; therefore, this condition warrants structured and regular screening [5].

The revised recommendations for HCQ toxicity screening issued by the American Academy of Ophthalmology and the clinical guidelines emanated by the Royal College of Ophthalmology suggest objective and structural screening tests, such as spectral domain optical coherence tomography (SD-OCT) and fundus autofluorescence (FAF) [5,6]. The rationale behind this is the absence of visible clinical signs of early toxicity but the ability of those imaging modalities to detect changes [7,8]. The first guideline also includes automated visual fields as a primary screening test [5], while the recommendations of the Royal College of Ophthalmology suggest that aside from SD-OCT and FAF, there is no solid evidence of any other structural exam in initial HCQ toxicity screening. However, if SD-OCT or FAF are abnormal, an investigation should be complemented with 10-2 perimetry and a multifocal electroretinogram (mfERG) if the visual fields are normal [6]. 

The long-term binding of HCQ to retinal pigment epithelium (RPE) cells has been hypothesized to be a plausible mechanism to explain toxicity [9]. Near-infrared reflectance (NIR) uses a longer wavelength (near-infrared) that penetrates the deeper layers of the retina, thus offering reflectance information at the level of the RPE, and these images are routinely acquired alongside SD-OCT imaging [10]. Therefore, NIR imaging has been suggested in the scarce literature that exists on the topic as a non-invasive method that may allow the detection of HCQ toxicity before ophthalmoscopic fundus changes or even before the loss of the ellipsoid zone on OCT [10,11]. Findings of HCQ toxicity described in NIR encompass speckled hyperreflectance at the center of the macula with an arcuate zone of hyporeflectance surrounding it, thus forming a bull’s eye appearance [11]. There is considerable controversy regarding the clinical meaning of these changes. Chew et al. [12] correlated this subclinical NIR bull’s eye lesion with the loss of the interdigitation zone in reflectivity maps using adaptive optics (AO) imaging. These regions were found to correspond to interdigitation-zone (outer segment tips) attenuation and loss of outer-segment cone signals. Interestingly, the area surrounding the bull’s eye lesion seen in NIR was interpreted as being the region of disease instead of the bull’s eye itself, which was further corroborated by microperimetry studies and with analysis of the anatomic correlation of the retinal cone mosaic on AO imaging [13]. In contrast, Wong et al. [10] commented that the demarcation line of the NIR bull’s eye lesion did not coincide with ellipsoid zone changes on OCT. Hence, biochemical changes related to early HCQ toxicity causing RPE abnormalities were presumed to be responsible for the increased reflectance.

This study aimed to provide further evidence of the role of near-infrared reflectance images in detecting early HCQ retinopathy using a large case series from a tertiary center.

## 2. Materials and Methods

### 2.1. Study Design

This retrospective cross-sectional study was approved by the local ethics committee at Centro Hospitalar Universitário de Lisboa Norte – Hospital de Santa Maria and conducted according to the principles of the Declaration of Helsinki. Informed consent was obtained.

### 2.2. Participants

Patients using HCQ who underwent routine retinal screening for HCQ toxicity at Centro Hospitalar Universitário de Lisboa Norte—Hospital de Santa Maria, Lisbon, Portugal from July to December 2021 were identified. 

Exclusion criteria included patients with incomplete clinical records and patients with media opacities or other causes of maculopathy that significantly precluded appropriate macular screening.

### 2.3. HCQ Screening Programme

All patients followed our Institution’s screening protocol for HCQ toxicity by performing SD-OCT (Heidelberg Engineering, Heidelberg, Germany), 30º FAF (Heidelberg Engineering, Heidelberg, Germany) and 10-2 test perimetry (Octopus 900, Haag–Streit, Köniz, Switzerland). Three clinicians reviewed the 3 exams as part of the routine virtual clinic (I.L., S.S.M. and P.J.) and decided if the examination was normal, altered but without need for HCQ suspension, or altered with the recommendation to withdrawal HCQ, and this information was collected.

### 2.4. NIR Imaging

Thirty degrees high-contrast digital NIR images (Heidelberg Engineering, Heidelberg, Germany) were captured using the 830 nm diode laser of the Spectralis System at the same time as the SD-OCT acquisition. A quality check was performed to confirm the appropriate focus and to exclude images with artifacts. NIR images were subsequently independently graded by 2 masked graders (M.S. and S.V.-P.) into normal, abnormal by the presence of an hyperreflective ring suggestive of a bull’s eye lesion, or abnormal by the presence of other fundus abnormalities (e.g., drusen, RPE changes, epiretinal membrane). In case of a grading disagreement, open adjudication was performed, and the images were reviewed by both graders at the same time to obtain the final grading.

### 2.5. Statistical Analysis

Statistical analysis was performed using Microsoft Office Excel 2021 for Mac version 16.54 and SPSS version 25.0 (IBM Corp, Armonk, NY, USA). Frequency and descriptive statistics were performed. Categorical variables were reported as a number or percentages, and continuous variables were recorded using mean ± standard deviation (SD). Normality testing was performed using Shapiro–Wilk’s test. A Chi-square or Fisher’s exact test was used to determine the associations between categorical variables. Continuous variables were compared using a Mann–Whitney U test or an independent samples T test, depending on the normality of the data. Measures of accuracy—sensitivity, specificity, positive predictive value (PPV) and negative predictive value (NPV)—were determined when appropriate. Concordance between graders for the assessed NIR images was determined with the Cohen’s k index. The strength of agreement was considered fair from 0.21 to 0.40, moderate from 0.41 to 0.60, substantial from 0.61 to 0.80 and almost perfect from 0.81 to 1.00, as previously described [14]. A *p*-value <0.05 was considered statistically significant.

## 3. Results

### 3.1. Clinical and Demographic Data

A total of 246 eyes from 123 patients were included, and 111 (90%) were female with a mean age of 55.2 ± 13.8 years. The most frequent rheumatic disease was SLE in 50 (41%) patients, followed by RA (*n* = 28, 23%) and primary Sjogren syndrome (pSS) (*n* = 19, 15%). The mean time of HCQ usage was 84.0 ± 72.3 months, and the mean weekly dose was 2327 ± 650 mg. The clinical and demographic data of the included participants is detailed in Table 1. 

Four patients were excluded due to the presence of significant macular disease such as cuticular drusen, best vitelliform macular dystrophy with choroidal neovascularization, sickle-cell retinopathy with macular involvement and *malattia leventinense*.

### 3.2. Results of routine HCQ Toxicity Screening

Of the 246 eyes screened, 4 (2%), 9 (4%) and 30 (12%) eyes were classified as abnormal on OCT, FAF and perimetry, respectively. Of these, one (0.4%) eye was classified as abnormal in all three exams, two eyes (0.8%) had an abnormal FAF with the rest of the exams being normal, and three eyes were classified as abnormal in both OCT and perimetry (no eye was classified as abnormal in just perimetry and FAF). Additionally, 53 (22%) of all perimetry exams were considered to have little clinical value due to low reliability index scores. This resulted in two patients (2%) being instructed to stop HCQ usage, although only one of these exhibited clear signs of HCQ toxicity, while the other was found to have a right-eye epiretinal membrane (ERM) (Figure 1a–d), which was considered to preclude adequate HCQ toxicity screening in the future (Figure 1e–h). Most patients (*n* = 113, 92%) were considered to have normal screening exams. A further five patients (5%) were considered to have an abnormal screening result but did not require HCQ usage termination. Eight (7%) patients had equivocal testing, mainly unreliable perimetry, and were thus scheduled for re-screening.

### 3.3. Near Infrared Imaging Results

In the retrospective analysis of NIR images obtained during routine HCQ screening, 175 of 246 eyes (71%) showed no changes in NIR reflectance, while 33 eyes (13%) were graded as exhibiting a hyperreflective ring suggestive of a bull’s eye lesion (Figure 2 and Figure 3). 

Thirty-six eyes (15%) showed other NIR abnormalities with varying and often non-specific characteristics.

### 3.4. Accuracy of NIR Imaging in Detecting HCQ Toxicity

The detection of eyes with HCQ toxicity as determined by the presence of a bull’s eye lesion using NIR imaging compared to a clinical decision based on OCT, FAF and perimetry is presented in Table 2. Two (0.8%) eyes with HCQ toxicity were identified in both NIR and primary routine screening. However, regarding the eyes showing a hyperreflective ring suggestive of a bull’s eye lesion, all 33 (13%) were found to have a normal SD-OCT, 4 (12%) had an abnormal FAF with a ring shape, and 19 (58%) had normal visual fields, while the remaining 14 (42%) had unreliable fields. The clinical decision for this subset of eyes was normal for 26 (81%), altered but without need for HCQ suspension for 2 (6%), and scheduled for re-screening in 4 (13%) because of unreliable visual fields.

The sensitivity for the detection of HCQ toxicity using NIR was 14%, with a specificity of 71%, PPV of 6% and NPV of 88%, respectively (Table 3). Fisher’s Exact test of independence was not statistically significant (*p* = 0.83). 

However, when performing a sub-analysis considering each of the imaging modalities, the Fisher’s Exact test was statistically significant (*p* = 0.04) when comparing NIR imaging to the reliable perimetry tests (193, 78%) and not statistically significant when comparing NIR imaging to OCT or FAF (respectively, *p* = 0.66 and *p* = 0.46). There was no statistically significant association between the NIR grading and the mean duration of HCQ usage (*p* = 0.30) or between the NIR grading and the mean dosage (*p* = 0.10).

### 3.5. Intergrader Agreement

The intergrader agreement for the NIR grading assessed was substantial, with a weighted kappa of 0.69 (standard error: 0.05, 95% confidence interval: 0.59–0.78). The graders disagreed in 34 of 246 eyes (14%). After open consultation of the 34 cases, 6 (18%) were confirmed as normal, 15 (44%) were confirmed as being abnormal by the presence of a hyperreflective ring, and 13 (38%) were revised as being abnormal by the presence of other fundus abnormalities that were not the presence of a hyperreflective ring.

## 4. Discussion

Due to the irreversible and possibly progressing nature of HCQ-induced RPE damage, detecting RPE abnormalities as early as possible is of paramount importance. 

This study aimed to increase the level of evidence from case reports and small case series [10,11,12] that indicated a possible role of NIR imaging in detecting early HCQ retinal toxicity by retrospectively analyzing a large cohort of patients with ongoing HCQ usage. With a mean duration of HCQ use of around 7 years and only one confirmed case of macular toxicity requiring treatment termination, the data presented here reaffirms once again the low incidence of HCQ toxicity, even with prolonged use. This case of bilateral macular toxicity was the only one to show abnormal results in all three primary routine screening exams.

In this analysis, we observed a large prevalence (33 eyes, 13%) of bull´s eye lesions identified on NIR, with a normal SD-OCT in all cases and with an abnormal FAF in just four eyes, suggesting that it may be an early sign of toxicity [10,11,12]. SD-OCT can detect early structural changes, and RPE dysfunction by increased lipofuscin accumulation is imaged by FAF [10]. However, there can be a disparity in the appearance of abnormalities in all these screening tests in the course of HCQ toxicity development [10,15]. Indeed, up to 10% of patients with the classic ring scotoma due to HCQ toxicity do not have evident structural SD-OCT changes [12,15]. In our series, we cannot comment on whether an NIR bull’s eye lesion arises from changes within or outside the interdigitation zone, as, in all cases, the SD-OCT was unremarkable [10,12]. However, in these cases, further testing may be of interest, such as using adaptive optics or the new cone mosaic metrics derived from Heidelberg Engineering Spectralis High Magnification Module images [12,13,16].

The high specificity and NPV values suggest that NIR imaging is a promising imaging modality for excluding HCQ toxicity. However, due to the low values of sensibility and PPV, it may not be an adequate imaging modality to diagnose or confirm the HCQ toxicity diagnosis. It may be employed as a first screening imaging method in the screening program, with the advantage that the NIR image is captured simultaneously with the OCT, requiring no additional testing time.

We have also found a substantial agreement between the two NIR reflectance image graders in identifying normal exams, those with a hyperreflective bull´s eye lesions, and those with other apparently unrelated abnormalities. This not only validates the presented findings of bull´s eye lesion prevalence but could also indicate an acceptable degree of inter-observer reproducibility, although a greater number of graders would have been preferable to assess reproducibility on a larger scale. The high intergrader agreement could also be suggestive of the ease with which experienced professionals could be trained to identify HCQ-related abnormalities in NIR images. Images were assessed effortlessly, and we agree with Wong et al. [10] that for the general ophthalmologist, it may be easier to recognize the bull’s eye maculopathy pattern compared to the outer retina paracentral thinning as observed in SD-OCT.

In our series, the absence of an NIR bull’s eye lesion only correlated significantly with perimetry but not with SD-OCT of FAF. However, all the reliable perimetry exams were normal and did not present an NIR bull’s eye lesion, which invalidates any association between the presence of an NIR bull’s eye lesion and abnormal perimetry. 

In fact, we found no association between the three exams and the presence of an NIR bull’s eye lesion, which may reinforce the idea that the ring is an early manifestation of HCQ toxicity [10,11,12]. It should also be noted that the use of perimetry as a primary exam for routine screening did not provide clinically valuable additional information in any of the cases when compared to SD-OCT and FAF. Additionally, apart from being a highly time-consuming and resource-consuming exam, the high rate of 22% of unreliable exams—results comparable to those presented by Marshall et al. [17] —makes its use questionable as a first-line exam. Thus, in line with the 2020 Royal College of Ophthalmology recommendations [6,18], the use of perimetry as a primary routine screening exam in our center has now been discontinued. 

Limitations include the retrospective nature of the study, as documenting the progression of an NIR hyperreflective bull´s eye lesion and observing changes over time in the other structural and/or functional exams is crucial to clarify the role of NIR imaging in HCQ toxicity screening. Thus, future longitudinal studies with a long-term follow-up are warranted to document NIR changes progressing to overt clinical toxicity. In this study, as in the other studies on this topic [10,11,12], we have used the Spectralis platform (Heidelberg Engineering, Heidelberg, Germany), which uses an 830 nm wavelength, so it remains unclear if these observations are reproducible using other commercially available devices.

In the future, artificially intelligent algorithms crossing data from various imaging modalities—SD-OCT, FAF, NIR and others currently under development—will probably constitute the future of HCQ-drug-toxicity detection.

## 5. Conclusions

In conclusion, we complemented previous anecdotal observations and provided evidence that a bull’s eye maculopathy pattern in an NIR image may correspond to early HCQ toxicity. Clinicians should be aware of this finding, and in the absence of abnormalities in other screening tests, an increased frequency of surveillance should be suggested.

## Figures and Tables

**Figure 1 diagnostics-13-00445-f001:**
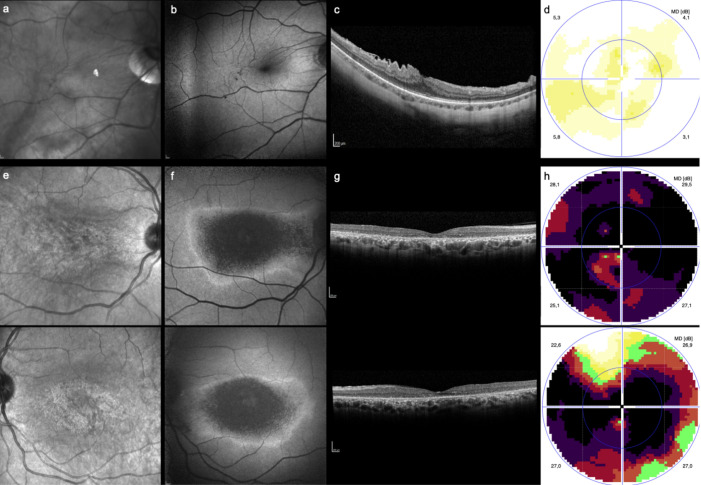
Exams of the two patients advised to stop HCQ usage. The first case related to a 60-year-old female using HCQ for 84 months (about 7 years). A right-eye epiretinal membrane was noted in the (**a**) NIR image, (**b**) FAF and (**c**) SD-OCT. The (**d**) 10-2 visual field showed a mild decrease of retinal sensitivity, and suspension of the drug was recommended. The second case (**e**–**h**) corresponds to a 59-year-old female using HCQ for 320 months (around 27 years) but with inconsistent screening, who was found to have bull’s eye maculopathy (right eye, top; left eye, bottom). Note the changes in both the (**e**) NIR image and (**b**) FAF and the advanced outer retinal atrophy in (**c**) SD-OCT. The (**h**) 10-2 visual field showed a bilateral central scotoma.

**Figure 2 diagnostics-13-00445-f002:**
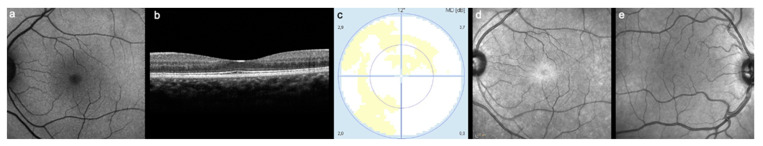
Case of a 40-year-old female taking a weekly dose of 2800 mg of HCQ for 36 months. The left eye shows an unremarkable (**a**) FAF, (**b**) SD-OCT and (**c**) perimetry. However, in the (**d**) NIR image (**d**), there are macular hyperreflective changes in accordance with a bull’s eye lesion. The (**e**) NIR of the right eye was normal.

**Figure 3 diagnostics-13-00445-f003:**
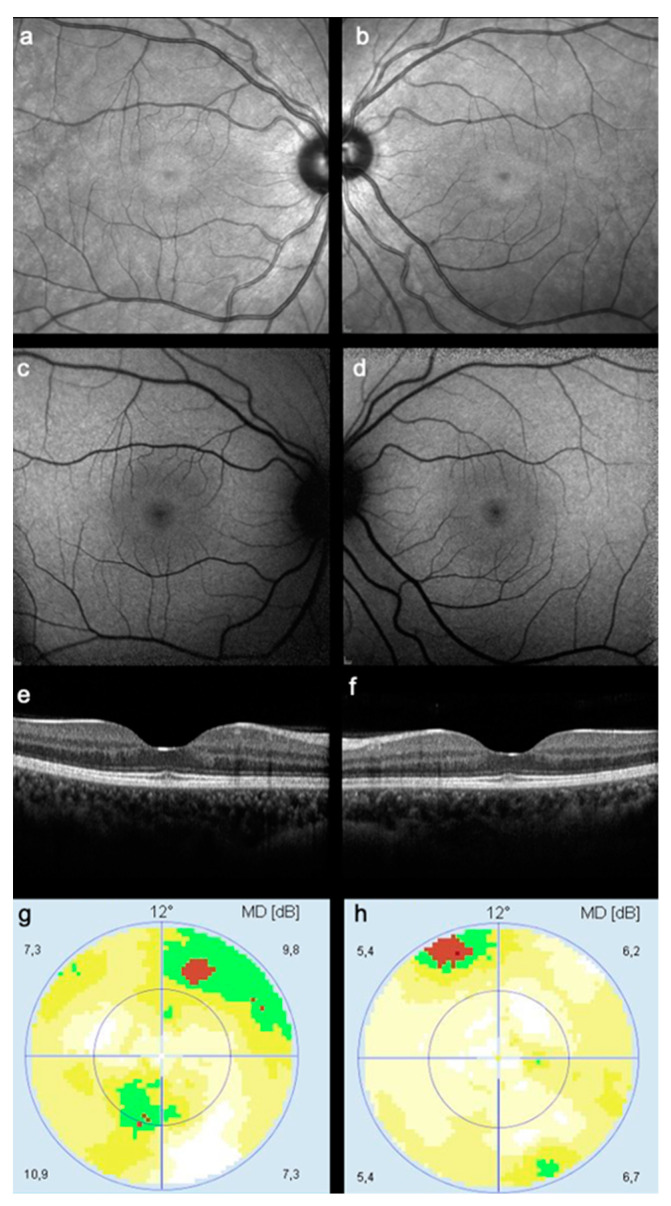
Case of a 23-year-old female with systemic lupus erythematosus using a weekly dose of 2800 mg of HCQ for 78 months. Note the presence of a hyperreflective ring in the NIR image suggestive of a bull’s eye lesion, in the (**a**) right and (**b**) left eye, respectively. The matching (**c**,**d**) FAF shows the paracentral hyperautofluorescence changes suggestive of a ring, but the (**e**,**f**) SD-OCT is unremarkable. The (**g**,**h**) 10-2 visual field testing was considered unreliable.

**Table 1 diagnostics-13-00445-t001:** Demographic and clinical characteristics of included patients.

Parameter	Results
Total *n* (eyes)	123 (246)
Mean Age ± SD, years	55.2 ± 13.8
Female *n* (%)	111 (90.2)
Primary Diagnosis *n* (%)	
RA	28 (22.8)
SLE	50 (40.7)
pSS	19 (15.4)
Other	24 (19.5)
Mean Time of HCQ Use ± SD, months	84.0 ± 72.3
Mean Weekly HCQ Dose ± SD, mg	2327 ± 650

HCQ = Hydroxycloroquine; pSS = Primary Sjögren Syndrome; RA = Rheumatoid Arthritis; SLE = Systemic Lupus Erythematosus.

**Table 2 diagnostics-13-00445-t002:** Detection of eyes with bull’s eye maculopathy using NIR imaging compared to result of the routine HCQ screening result.

	Result of the Routine HCQ Screening			
NIR Imaging	Positive, *n* (%)	Negative, *n* (%)	Unsure/Ungradable, *n* (%)	Total, *n* (%)
Positive, *n* (%)	2 (0.8)	27 (11.0)	4 (1.6)	33 (13.4)
Negative, *n* (%)	11 (4.5)	154 (62.6)	10 (4.1)	175 (71.1)
Unsure/ungradable, *n* (%)	1 (0.4)	35 (14.2)	2 (0.8)	38 (15.4)
Total, *n* (%)	14 (5.7)	216 (87.8)	16 (6.5)	246 (100)

**Table 3 diagnostics-13-00445-t003:** Sensitivity and specificity of NIR imaging in detecting bull’s eye maculopathy.

NIR	Sensitivity (%)	Specificity (%)	PPV (%)	NPV (%)
HCQ toxicity	14.3	71.3	6.1	88.0

PPV = Positive Predictive Value; NPV = Negative Predictive Value.

## Data Availability

The data is available from the corresponding author upon reasonable request.

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
