# Peer review of "Bull’s Eye Maculopathy in Near-Infrared Reflectance as An Early Sign of Hydroxychloroquine Toxicity"

_diagnostics, 2023, doi:10.3390/diagnostics13030445_

Round 1
Reviewer 1 Report
In this article, Santos et al present a retrospective study on individuals on Hydroxychloroquine (HCQ) regimen to assess any changes in their retinal exams that may be indicative of HCQ retinal toxicity. The authors have based their study on previous work published by Wong et al where near-infrared reflectance (NIR) was proposed to be an early indicator of HCQ toxicity. Identification of such early indicators are important because maculopathy caused by HCQ toxicity, although rare, is devastating for affected individuals. To be clear, the authors have not identified the approach but expanded on a previous study. Such follow-up studies are still important to provide relevant datasets to the field, and to update HCQ dosing regimen to prevent vision loss in patients who might be at risk of developing Bull’s eye maculopathy.
Below are the comments on the manuscript:
1. Is there NIR data available for patients who have Bull’s eye maculopathy and for the two patients who were recommended to stop HCQ? Please include those images, and also mention which patient clearly indicated signs of HCQ toxicity and which one did not but was recommended to discontinue HCQ treatment.
2. Figure 1: The authors mentioned that the right eye of the patient was normal. It would be helpful to have images for both left eye and the right eye to allow comparison for readers, especially if the presence of hyperreflective changes in left eye can be compared to absence of such changes in the right eye.
3. Intergrader agreement is an important part to suggest routine use of NIR to assess HCQ toxicity. However, the authors only provide one sentence in this results section. Please use numbers and percentages that matched between the graders for assessment. Please also provide numers for cases/images where open consultation was used because of grader disagreements. What would make this study more useful is if the authors could include a third grader if possible.
4. Line 36: It would be better if instead of ‘several dermatological conditions’, the authors could state ‘several dermatological conditions such as…..’ and provide a couple of examples.
5. Lines 23-24 and lines 127-128 are exactly the same. Please update so the text does not exactly match but the overall content is same.
6. Line 130: It is not clear why are there two numbers are written in parentheses. Does it refer to female and male patients? If it is meant as patient numbers and %, please use (n=X; Y%) format for clarity.
Reviewer 2 Report
It was an interesting paper, with a fresh perspective on improving early screening of macular toxicity due to HCQ. The paper is well written and concise. What I would have improved is the number of statistically significant cases, given that the study cases included mostly (92%) normal screening exams. Authors have included four screening methods: OCT, FAF, 10-2 perimetry and NIR, but as the authors noted, these do not have high intercorrelation. There was a high variation between the percentage of abnormal eyes between these methods, with only 0.4% eyes being classified as abnormal in OCT, FAF and perimetry.
On page 2, line 56, RPE should be defined, seeing how it's the first time it appears. Moreover, that is not quite whta comes out of ref. 9, but rather that despite symptoms, OCT images in that case study did not show any changes in the RPE, which relates to the next sentence of this paragraph, that better results might be obtained with NIR, which penetrates deeper into the tissue.
Page 3, lines 129-130, should include the abbreviations of all the diseases, and these must be removed from the end of Table 1.
Page 6, Tables 2-3, abbreviations would be better removed from the end of the tables and included in the text.
Overall, I believe that this is an interesting study, which shows promising results but must be enlarged.
